# IFFMStyle: High-Quality Image Style Transfer Using Invalid Feature Filter Modules

**DOI:** 10.3390/s22166134

**Published:** 2022-08-16

**Authors:** Zhijie Xu, Liyan Hou, Jianqin Zhang

**Affiliations:** 1School of Science, Beijing University of Civil Engineering and Architecture, Beijing 102616, China; 2School of Geomatics and Urban Spatial Informatics, Beijing University of Civil Engineering and Architecture, Beijing 102616, China

**Keywords:** style collection, IFFM, semantic style transfer

## Abstract

Image style transfer is a challenging problem in computer vision which aims at rendering an image into different styles. A lot of progress has been made to transfer the style of one painting of a representative artist in real time, whereas less attention has been focused on transferring an artist’s style from a collection of his paintings. This task requests capturing the artist’s precise style from his painting collection. Existing methods did not pay more attention on the possible disruption of original content details and image structures by texture elements and noises, which leads to the structure deformation or edge blurring of the generated images. To address this problem, we propose IFFMStyle, a high-quality image style transfer framework. Specifically, we introduce invalid feature filtering modules (IFFM) to the encoder–decoder architecture to filter the content-independent features in the original image and the generated image. Then, the content-consistency constraint is used to enhance the model’s content-preserving capability. We also introduce style perception consistency loss to jointly train a network with content loss and adversarial loss to maintain the distinction of different semantic content in the generated image. Additionally, we have no requirement for paired content image and style image. The experimental results show that the stylized image generated by the proposed method significantly improves the quality of the generated images, and can realize the style transfer based on the semantic information of the content image. Compared with the advanced method, our method is more favored by users.

## 1. Introduction

The process of separating and recombining content and style of images using neural representations is called neural style transfer. Gatys et al. [1] first proposed this technique and successfully created high-quality artistic images using a convolutional neural network. Since then, neural style transfer had drawn much attention in the field of computer vision. However, based on image optimization, the speed is limited. Therefore, researchers have proposed many different algorithms for accelerating the realization of style transfer.

Methods in [2,3,4] introduce a feedforward network to accelerate the process of style reconstruction. These methods can perform real-time single style transfer. However, for transferring new style, it is needed to retrain the model. some studies [5,6] propose embedding affine transformation in the middle of the autoencoder to achieve style transfer of multiple styles. To transfer new styles, the methods only need to retrain the middle layers separately instead of retraining the entire model. Anyway, the above methods take extra time to retrain the model when transferring a new style. Furthermore, methods [7,8,9,10] propose style feature embedding networks to achieve arbitrary style transfer without retraining the model.

The zero-shot methods mentioned above [7,8,9,10] can only transfer the style of one painting of representative artist, which is not enough to represent the artist’s general artistic style. Recently, GAN-based style transfer methods are proposed to transfer artist’s style from the image collection [11,12,13]. A content image can be transferred to a kind of style of a painter instead of a single style artwork. The style transfer methods based on the style collection integrate the collective characteristic of the images, and it is essentially fusing a content image with multiple style images of similar themes. Complex textures in so many style images may dilute the structure and content of the original image. The generated images are prone to distortion or edge blurring. (See Section 4.3).

In this paper, we propose to embed a pair of invalid feature filter modules in GAN to filter out the secondary features not related to the structure and to better preserve original content features and local structures. In addition, it is important to perform style transfer according to semantic information. This can help distinguish different semantic content in the stylized images. Therefore, we introduce the style perception loss function to measure the difference between the style image and the stylized image in the latent space. By jointly optimizing the style loss function and the content loss, the output images are encouraged to transfer the corresponding style according to the semantic features of the content. Experiments show that proposed method can achieve semantic-related style transfer. As shown in Figure 1, the part in the red box reflects that our method can better retain the green part of the lawn in the original image, and we turn the color of the sky and lake into blue.

It should be noted that our method does not have the requirement for paired data, and does not need to manually select the content image matching the style image for training as in [11,13]. For constructing the collection of style images, given a style image, we automatically select the style images related to given image from the style image dataset, and the content image can be any photo. Comparison experiments and ablation experiments show that our method can obtain better stylized images, and can achieve style transfer according to the semantic knowledge of the content, which also significantly improves the texture distortion and uneven color distribution.

The main contributions of this paper are: (1) We introduce a pair of invalid feature filter modules in the network to better preserve the structural features of the content. (2) We propose a high-quality style transfer method based on style collection, which represents a kind of style of a painter. By jointly training the style perception loss and content feature loss with a GAN architecture, the style transfer based on semantic information can be achieved. (3) Experiments show that the stylized results generated by our method can meet the needs of high-quality and real-time style transfer.

## 2. Related Work

### 2.1. Style Transfer

Early style transfer algorithms include methods [14,15] based on image rendering that belongs to the field of non-photorealistic graphics. However, these traditional methods have not been applied to the industrial field on a large scale due to restrictions such as a single type of style or must be paired input. The style transfer algorithm in the computer vision field includes the method of texture transfer [2,16]. This type of method does not consider the semantic relationship between content and style and only transfers low-level image features, so the result of stylization is not very satisfactory.

Gatys et al. [1] proposed the use of convolutional neural networks for style modeling in style images. The stylization results obtained by this method are very effective, and for the first time the idea of deep learning is used for style transfer, which has laid a good foundation for the development of style transfer. In recent years, there has been an endless stream of research on neural style transfer. Wang et al. [17] proved that style transfer can be regarded as a domain adaptation problem through derivation based on [1]. Li et al. [16] introduced Laplacian loss to guide image synthesis, and Li et al. [18] proposed using Markov random field loss instead of Gram matrix loss in order to improve the style characteristics of style transfer.

The above methods based on model iteration have great limitations in speed, which are not convenient for industrial applications. In order to solve the speed problem, Ulyanov et al. [3] used a multi-scale texture network to synthesize stylized images, on this basis, Ulyanov et al. [4] set the batch size to 1 and introduce instance normalization layer training to make the model converge faster. Johnson et al. [2] proposed to train a forward residual model with perceptual loss. Two perceptual loss functions are defined to measure the high-level perception and semantic differences between images, which can achieve style transfer based on semantic knowledge. However, this type of methods can only realize the transfer of one style image for one content image by training a model, so multi-style and arbitrary-style style transfer methods are proposed.

Chen et al. [5] proposed that StyleBank is bound to each style so that only this part needs to be retrained when converting a new style image. Dumoulin et al. [6] proposed conditional instance normalization. To transfer to the new style, just do an affine transformation on the instance normalization layer. The above methods still need extra time training to achieve the style transfer of the new style, so Li et al. [8] proposed the use of whitening and coloring operations plus the structure of the encoder and decoder to achieve arbitrary style transfer. Xun et al. [7] proposed adaptive instance normalization to directly normalize the content in the image into different styles through large-scale training; Sheng et al. [2] proposed a style decorator to align style features to images corresponding to semantic information to achieve multi-scale zero style transfer. Park et al. [9] proposed a style attention network and identity loss function to achieve arbitrary real-time transfer.

Recently, there are several methods [11,12,13] based on generative adversarial networks [19] that can achieve the style transfer of a class of style collection. The generative adversarial network is composed of two parts: generator and discriminator. The main thought is mixing the spurious with the genuine to achieve a dynamic balance. Zhu et al. [13] introduced the cycle consistency loss to transfer the image from the source domain to the target domain, but the result generated by this method will produce rough texture because the loss function measures the difference between the generated image and the content image in the RGB space. In order to solve the above problems, Sanakoyeu et al. [12] proposed a style-aware content loss function to optimize the content features in the stylized results, but the stylized images may inevitably produce deformations and texture distortions. Ma et al. [11] proposed that [12] did not consider semantic information and regarded the content domain and style domain as two separable parts, so they proposed dual consistency loss to achieve semantic-related style transfer. However, the limitation lies in the need to manually select content images related to the theme of the style image. Recent contrastive learning models [20,21,22] show that the non-linear projection head can filter out the invalid features. Therefore, this article refers to the idea of introducing an invalid feature filtering module into the encoder–decoder structure, which extracts the structural features of the content after the input image and the generated image are filtered to avoid the interference of redundant texture features.

### 2.2. MLP

After Ting Chen of the Hinton group proposed SimCLR [20], contrastive learning has attracted strong attention. Subsequently, many scholars have proposed different comparative learning models [21,22,23], and their models even exceed the performance of supervised learning. The common point of these models is that they all introduce Non-linear projection head behind the encoder.

The study of SimSiam [23] also shows that its performance is hardly improved without Non-linear projection head. The main function of MLP is to filter out the invalid feature that is represented to obtain the essence. Additionally, we just want to avoid the interference of redundant texture features, so we introduce an invalid feature filtering module into the model, which is mainly composed of Non-linear projection head.

Recently, Non-linear projection head has set off a wave of enthusiasm in the field of computer vision, because the Google team, the Tsinghua team, and scholars from Oxford University [24,25,26,27] introduced the use of Non-linear projection head-based models to achieve image classification, semantic segmentation, image generation and other visual tasks. Its experiments also show that its efficiency and accuracy can achieve comparable effects with convolutional neural networks and transformer modules. This also further proves the rationality of our application of the Non-linear projection head module, and our experimental results also confirm that the stylization results have been significantly improved.

## 3. Method

### 3.1. Network Architecture

The network architecture of our method is shown in Figure 2. It is a GAN-based approach. The generator is composed of an encoder and decoder structure. The encoder E extracts the features of the input image $x_{c}$ and maps it to the representation space $z=E(x_{c})$. The decoder G is used to generate the stylized image $x_{cs}=G(z)$. The discriminator D(·) is used to distinguish the generated stylized image $x_{cs}$ from the real original style image $x_{s}$. Specially, we embed invalid feature filtering module I in the network to optimize the content structure of the generated image.

### 3.2. Training

We use the standard discriminator loss function to optimize the style characteristics of the generated results. The adversarial loss is as follows:(1)$$\begin{array}{l} L_{adv}(E,G,D)=E_{y\sim p_{Y}(y)}[\log D(y)]\\ \;\;\;\;\;\;\;\;\;\;\;\;\;\;\;\;\;\;\;\;\;\;\;\;+E_{x\sim p_{X}(x)}[\log(1-D(G(E(x)))] \end{array}$$
where *y* represents a style image, y∈Y, x∈X, and *x* represents a content image.

In addition to capturing style from style images, preserving the content structure of the original image is important. We consequently introduce the content-consistency constraint to enhance the model’s content-preserving capability. The methods based on the style collection combine the collective characteristic of the images. Complex textures in so many style images may dilute the structure and content of the original image. The generated images are prone to distortion or edge blurring. In order to reduce the interference of the texture unrelated to the structure, we propose to introduce invalid feature filter modules (IFFM) into the network to filter out the invalid features in the image. We define the content loss between content image $x_{c}$ and the output image $x_{cs}$ as: (2)$$L_{content}(E,G)=\frac{1}{CHW}∥I(x_{c})-I(x_{cs}){∥}_{2}^{2}$$
where CHW is the size of the input image $x_{c}$, I is invalid feature filtering module. The detailed structure of IFFM is described in Section 3.2.

In addition, performing style transfer according to different semantic content is significant. Therefore, we introduce style perception loss to measure the difference between style image $x_{s}$ and stylized image $x_{cs}=G(E(x_{c}))$ in latent space. In this way, the generated image preserves desired content information of the original image according to corresponding style. The style perception loss and content loss are jointly trained with GAN to optimize semantic features of stylized images. However, directly using the Gram matrix to match the pixels between the style image and the generated image may cause the generated result to appear messy and uneven color distribution, so we define style perception loss as the Euclidean distance between generated stylized image $x_{cs}$ and real style image $x_{s}$ in latent space:(3)$$L_{style}(E,G)=E_{y\sim p_{Y}(y)}[\frac{1}{d}∥E(y)-E(G(E(x))){∥}_{2}^{2}]$$
where *d* is the dimension of the latent space.

Using the above three losses, the total loss is formulated as:(4)$$L(E,G,D)=L_{content}+L_{style}+\lambda L_{adv}$$
where λ is the weight parameter that controls content style consistency loss and adversarial loss. We optimize our model through the following optimization problems.
(5)$$E,G=\arg\;\underset{E,G}{\min}\;\underset{D}{\max}L(E,G,D)$$

### 3.3. Invalid Feature Filtering Module

Our proposed invalid feature filtering module (IFFM) consists of a pair of non-linear projection heads h(·), which are used to filter out the redundant texture features of the input image and the generated stylized image. Since our style transfer algorithm is based on a collection of style images, generated stylized images may contain rich textures of multiple style images, which may interfere with the content structure. Therefore, we use the invalid feature filtering module to only retain the structural features related to the content and then measure the content loss, which can reduce the occurrence of edge distortion and structural deformation. $h(\cdot )=W^{\left(2\right)}\sigma (W^{\left(1\right)}x)$ is obtained through projection head modules with two layers, where σ is a RELU non-linear activation function [20].

The reason why we thought of using non-linear projection heads for filtering is because Chen et al. [23] introduced non-linear projection heads into the network, and showed through experiments that the prediction accuracy performance of the model without the projection head was poor. Additionally, it reached the same conclusion as [20]: the projection head module plays an important role in removing the invalid feature of the image.

However, unlike the classic MLP structure, we replace the batch normalization layer with the instance normalization layer. The instance normalization is suitable for the generation model, especially the style transfer, so applying the instance normalization layer not only allows the model to converge faster, but also maintains the independence between images [28]. This prevents instance-specific mean and covariance shift simplifying the learning process. Differently from batch normalization, furthermore, the instance normalization layer is applied at test time as well. The normalization process allows to remove instance-specific contrast information from the content image, which simplifies generation. The structure of the invalid feature filtering module is shown in Figure 3.

The idea of [18]’s method is similar with ours. Sanakoyeu et al. [12] proposed to inject a transformer block to the model, then measured transformation loss to discard unnecessary details in the content image according to the style. The image transformation is achieved through a pooling layer, which may cause the content image to become fuzzy instead of filtering out the true redundant features of the image, so the resulting image may be deformed more severely.

## 4. Experiments

### 4.1. Training Details

The basic model of our encoder decoder adopts the structure in [2]. The encoder network consists of 5 convolutional layers: 1 convolutional block with a step length of 1 followed by 4 stride-2 convolutional block, and the decoder network contains 9 residual blocks [29] that are composed of 4 up-sampling blocks and a convolutional layer with stride-1. In addition, the instance normalization layer is used after the convolutional layer. The discriminator network uses the multi-scale structure from [30] that is a fully convolutional network consisting of 7 convolutional blocks with stride-2. For the training of the overall network, we set λ in Equation (4) to 0.01. We train for a total of 300,000 iterations and the batch size is 1. The Adam [30] optimizer is used and the learning rate is set to 0.0002 to optimize the network.

### 4.2. Data Composition

We use Places365 [31] as the dataset of content images in order to better adapt to the style transfer of more scenes. This dataset contains 365 scene classes in life, covering a wide range and comprehensiveness, including 1.8 million training images. We randomly sampled 768 × 768 image patches from the dataset as content images for training. There is no need to manually choose the paired images related to the structure theme as in [24]. We adopt the collection of style images selected by automatic grouping in [26] from Wikiart [32]. An example of the style image collection after style grouping is shown in Figure 4 (part of the Van Gogh style image collection example).

### 4.3. Experimental Results

We compared our method with the previous three types of works, including collection-based methods [12,13], arbitrary style methods [7,8] and single style transfer methods [1,17]. For [7,8], we use the pre-trained model provided by the author; For methods [12,13] and [1,17], we use the source code provided by the author to train through our dataset.

Figure 5, Figure 6 and Figure 7, respectively, shows three style collections, which are style of Van Gogh, Picasso and Monet. Our method is compared with CycleGAN [13] and the style-aware content loss method [12]. The results of method [13] show that it is difficult to distinguish different content because the colors of different content become similar. For example, the lake in the third and fifth rows of Figure 5 has been integrated with stones and bridges and the color all turned green, while the color of lake in the stylized image generated by our method has turned blue, and the color of tree trunks and stones are brown. Our method can clearly distinguish the colors of different content. The fundamental reason why the stylized results generated by CycleGAN [13] are not so natural is that the cycle consistency loss directly measures the difference between the stylized results and the reverse mapping of the content images in the RGB space. This has a bad influence on the results for content images and style images with large structural differences.

For Picasso style, the structure of the style image is relatively abstract and the process of style transfer is decomposed and reassembled, so the difference in the structure of the content image and the style image will not have a particularly large impact on generated results. However, the stylization result of [18] may show blurred edges and uneven color distribution (see line 6 in Figure 6). For Monet’s Impressionist style, the stylized results produced by CycleGAN may produce noisy textures and brushstrokes (see lines 1, 4, and 5 in Figure 7).

In order to solve the above problems, Sanakoyeu et al. [12] proposed the style-aware content loss to determine the content details to be retained according to the style. This may result in serious structural deformation. For example, in the fourth row of Figure 6, the woman’s legs have been merged with the background and the pole behind her is also severely deformed. In comparison, our method better weighs the representation between content images and style images, and better retains the structure of content features relatively. In addition, our method can achieve style transfer based on different semantic content. The method [12] cannot perform style transfer based on the semantic relationship between the content image and the style image. For example, in lines 3, 5, and 6 in Figure 5, our method can convert color of lake to blue and color of tree trunks, stones and bridges to brown, but the method [12] cannot do this.

We also compared with fast arbitrary style transfer methods, including WCT [8] and AdaIN [7]. Figure 8 shows the comparison results of our method and two zero-shot style transfer methods. These two methods can retain good style characteristics, and do not limit the types of styles. However, in the results generated by the method of WCT [8], more content features are lost, and style features are over reserved. The stylized results generated by AdaIN [7] are more prone to texture bending and distortion. In contrast, our method can better retain the content characteristics and it is not easy to appear deform the structure.

Figure 9 shows the comparison between our method and the researches based on single style transfer. In theory, the effect of using a convolutional neural network to achieve style transfer proposed by Gatys et al. [1] is the best. However, as shown in Figure 9, the methods [1,17] implement style transfer between a single content image and a single style image, which may cause uneven color distribution. On the contrary, our stylized images can represent a type of painting from a certain painter. Therefore, our stylized results naturally contain the information of multiple style images, and the colors of the styles corresponding to different content in our generated images are also different, which greatly reduces the uneven color distribution. Although the studies [11,12] mentioned that the transfer based on the style image collection can be realized by calculating the Gram matrix of different style images, it also shows that the effect of the style transfer based on the calculation of the Gram average is not good. Our method can not only preserve the visually reasonable results, but also meet the real-time style transfer.

In addition, the details of our stylized images are kept clear when the size of the stylized result generated by our method is 1280 × 1280 pixels, which means that our method can meet the needs of high-definition transfer images. As shown in Figure 10 the images generated by our method can retain fine details and strokes even with high resolution.

### 4.4. User Research

There is currently no more authoritative unified quantitative evaluation standard to judge the performance of style transfer tasks. Moreover, the goal of style transfer is to meet the needs of users, so the subjective feelings of users are crucial to the evaluation of stylized results. Therefore, we evaluate different methods through user voting. We compare our method with three researches [7,12,13]. For each method, we choose three styles, and randomly select 5 style transfer images for each style. Given the original content image and the corresponding style, users choose their favorite image from the images in each style. Figure 11 shows the result of collecting 1230 votes from 82 users and converting them into an average percentage. It is obvious that our method is more popular.

We also count the percentage of votes obtained by the three styles of each method. As shown in Figure 12, our method IFFMStyle comprehensively got the most votes among the three styles. However, the number of votes in the styles of Picasso and Monet is slightly less than the method of AdaIN [7]. The reason is that the image structure of Picasso style and Monet’s style is more complicated, so the images change larger after style transfer, and users are more inclined to choose a result that is closer to the original content image. Our method is to synthesize multiple styles of a painter, while AdaIN’s stylized images represent only single style of stylized image, which leads to slight differences in the generated results. Moreover, the users may be less sensitive to distortion of the stylized image. For the Van Gogh style, most users choose our stylized images. Overall, our method can better meet the needs and preferences of users.

### 4.5. Ablation Experiment

#### 4.5.1. Loss Function

We produce ablation experiments to verify the effectiveness of each loss function, as shown in Figure 13. The content structure loss in the generated results is very serious because only the adversarial loss is used to train our network (b). The color distribution of the results generated without content loss training is not natural (c). Instead of style loss to train the network, the content structure of generated images is unclear and the texture features are disordered (d). We use auto-encoder loss to replace the content loss of the IFFM, which result in the content features not being retained well (e). The stylized image generated by training the network with the above three loss functions is optimal (f). This also proves that every piece of our model is meaningful, and the performance is best when all the modules are combined.

#### 4.5.2. Analysis of Weight Parameters

We also analyze the influence of the weight parameter in Equation (4) on the experimental results. The weight parameter λ mainly controls the importance of adversarial loss, and adversarial loss mainly optimizes style characteristics, so the style characteristics become more and more obvious with the increase in λ in Figure 14. In order to better balance content features and style features, we set λ to 0.01 in a comprehensive consideration.

## 5. Conclusions

We propose a real-time style transfer method based on a collection of style images, which can achieve style transfer according to the semantic information of content images. There is no pairing restriction on the content image and the style image. In addition, we propose the invalid feature filtering modules in the encoder decoder structure to filter the redundant feature of the input images and the generated images, which can reduce the interference of features not related to structure. Style transfer based on semantic features can be achieved to alleviate the color disorder by jointly training style-content-consistency loss and adversarial loss. Experiments prove that the quality of the stylized results is high and can meet the needs of high definition and real time, and the images generated by our method are more popular than other advanced methods.

The main limitation of this work is that there is no unified objective standard for the evaluation of stylized images, and it can only rely on people’s subjective judgments, so different people may have inconsistent preferences. In addition, the content images targeted by our method are various scenes in life, and the styles are paintings of many painters. If a new style is transferred to a specific image, the effect achieved by this method may not be applicable.

## Figures and Tables

**Figure 1 sensors-22-06134-f001:**
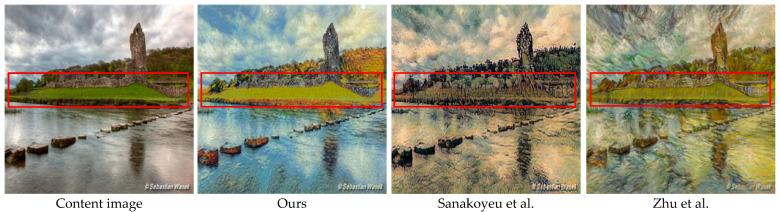
Comparison with Sanakoyeu et al. [12] and Zhu et al. [13] from Van Gogh’s style. These methods implement style transfer based on a collection of style images. The red frame shows that the effectiveness of jointly training IFFM content loss and style loss, and our method can achieve style transfer based on semantic information.

**Figure 2 sensors-22-06134-f002:**
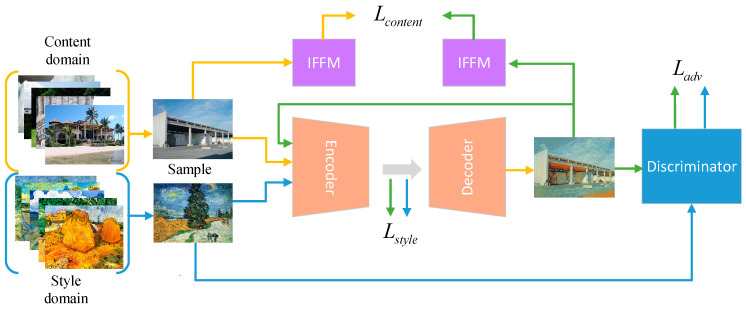
Network architecture.

**Figure 3 sensors-22-06134-f003:**
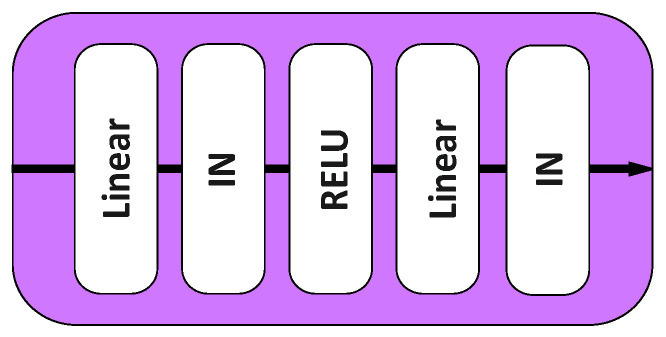
IFFM architecture.

**Figure 4 sensors-22-06134-f004:**
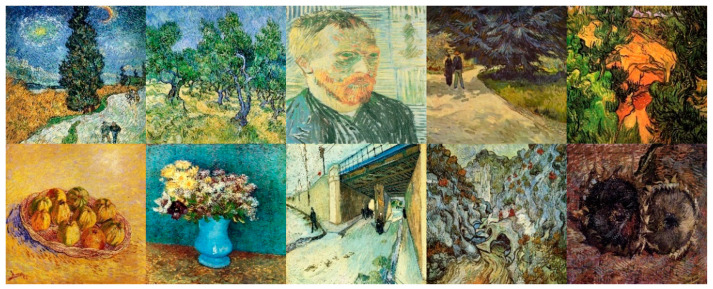
Examples of Van Gogh style image collections.

**Figure 5 sensors-22-06134-f005:**
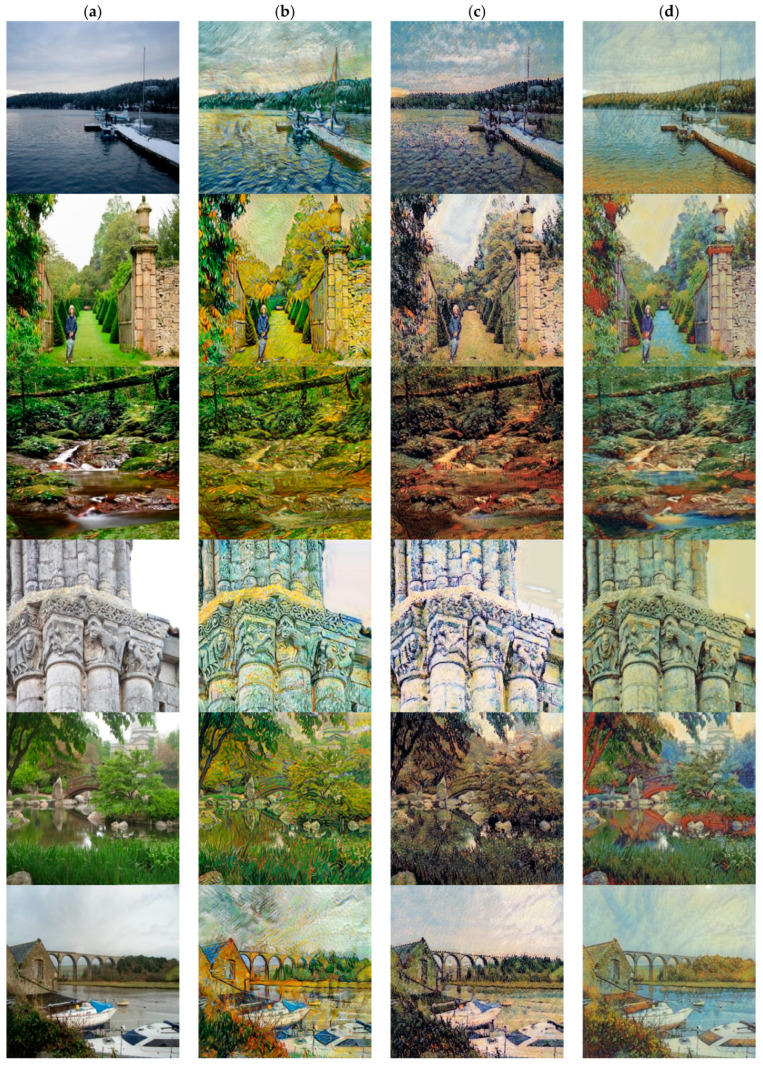
Comparison with different style transfer methods based on the Van Gogh style collection. (**a**) Inputs. (**b**) CycleGAN [13]. (**c**) Sanakoyeu et al. [12]. (**d**) Ours.

**Figure 6 sensors-22-06134-f006:**
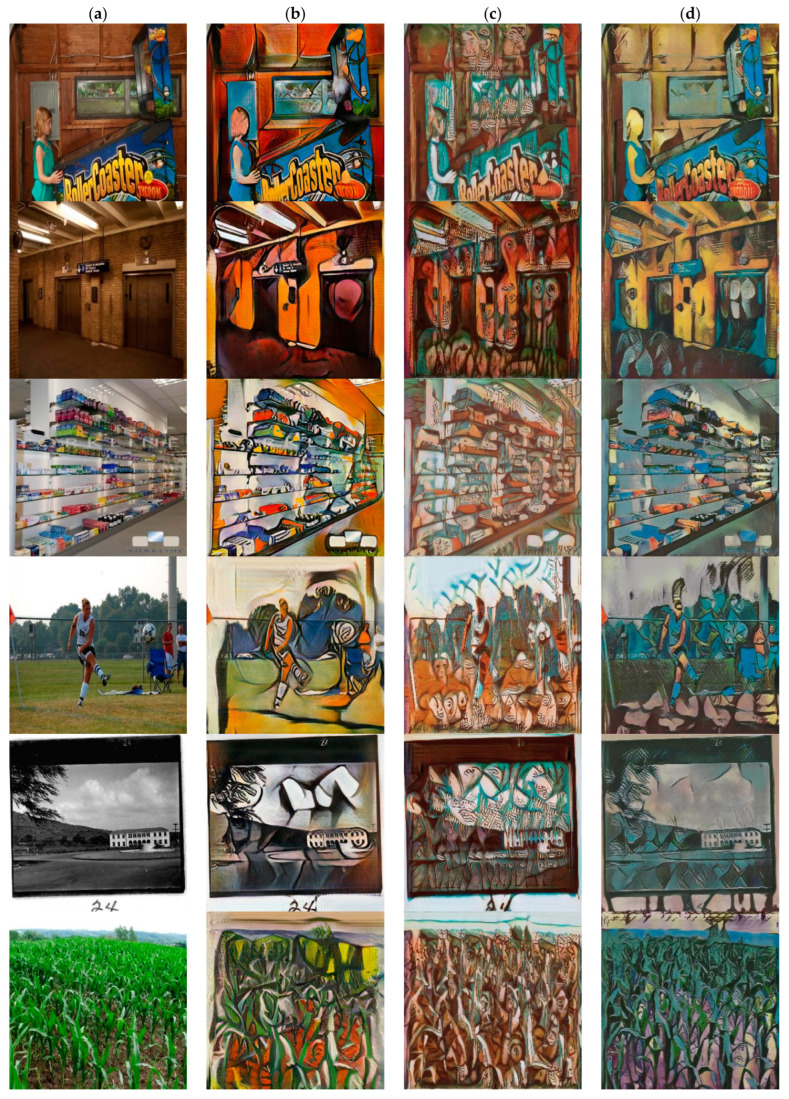
Comparison with different style transfer method based on the Picasso style collection. (**a**) Inputs. (**b**) CycleGAN [13]. (**c**) Sanakoyeu et al. [12]. (**d**) Ours.

**Figure 7 sensors-22-06134-f007:**
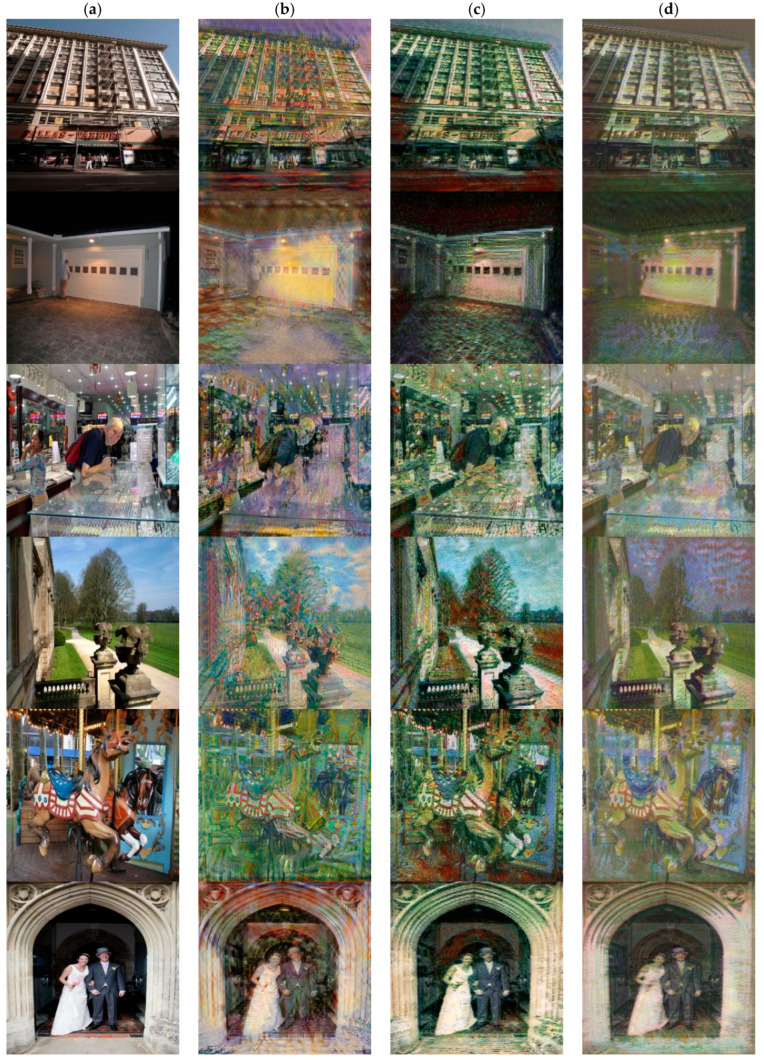
Comparison with different style transfer method based on the Monet style collection. (**a**) Inputs. (**b**) CycleGAN [13]. (**c**) Sanakoyeu et al. [12]. (**d**) Ours.

**Figure 8 sensors-22-06134-f008:**
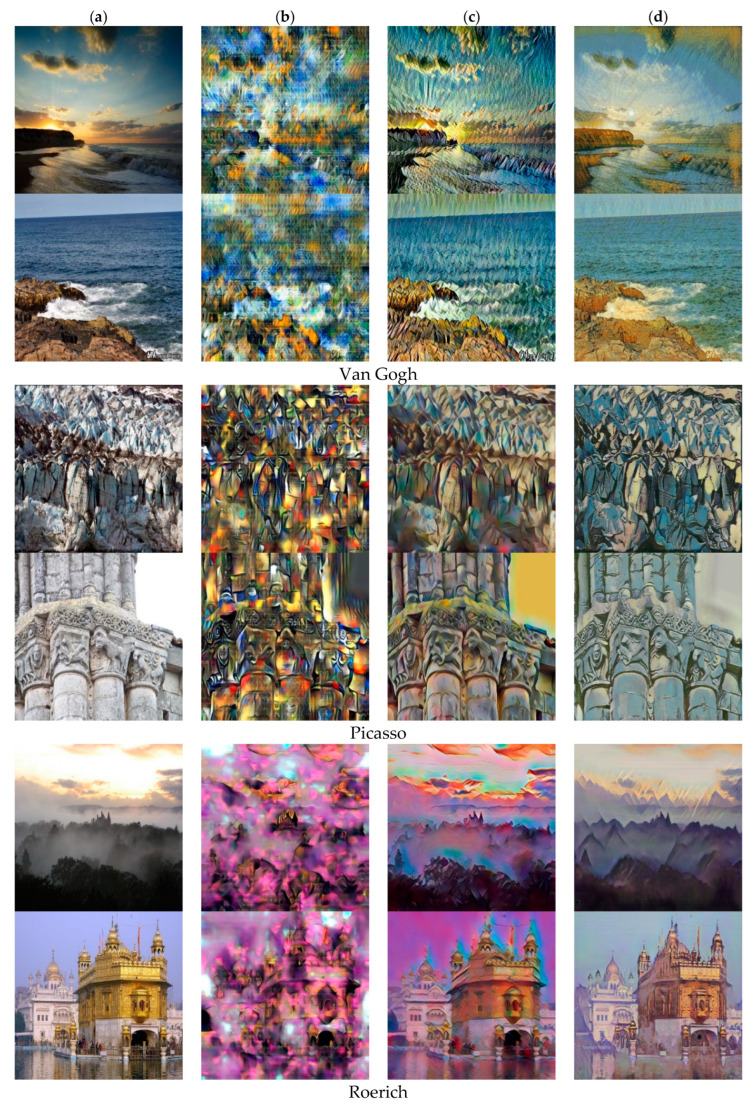
Comparison with the three styles of the zero-shot methods. (**a**) Inputs. (**b**) WCT [8]. (**c**) AdaIN [7]. (**d**) Ours.

**Figure 9 sensors-22-06134-f009:**
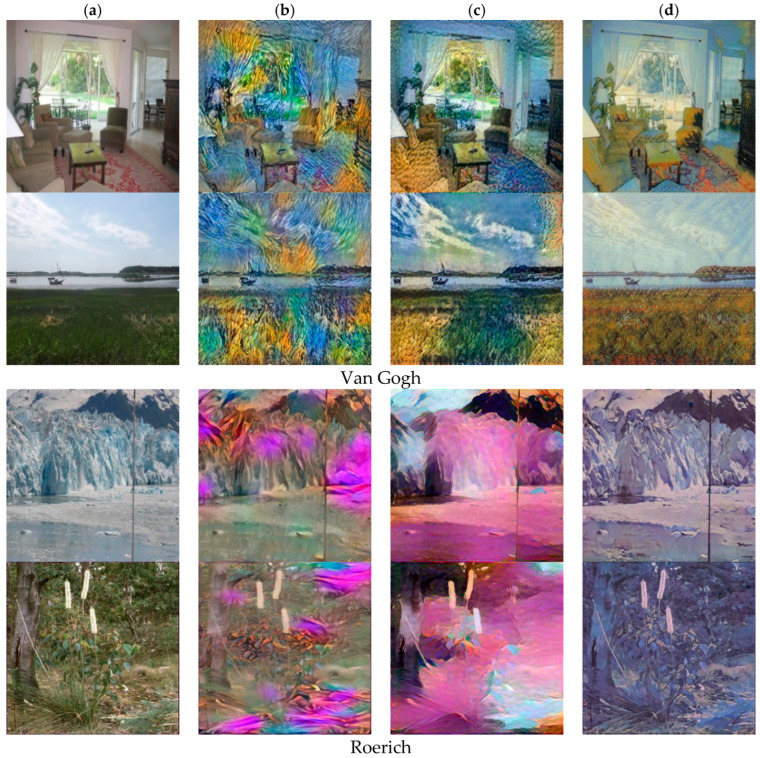
Contrast with slow style transfer methods. (**a**) Inputs. (**b**) Gatyes et al. [1]. (**c**) MMD [17]. (**d**) Ours.

**Figure 10 sensors-22-06134-f010:**
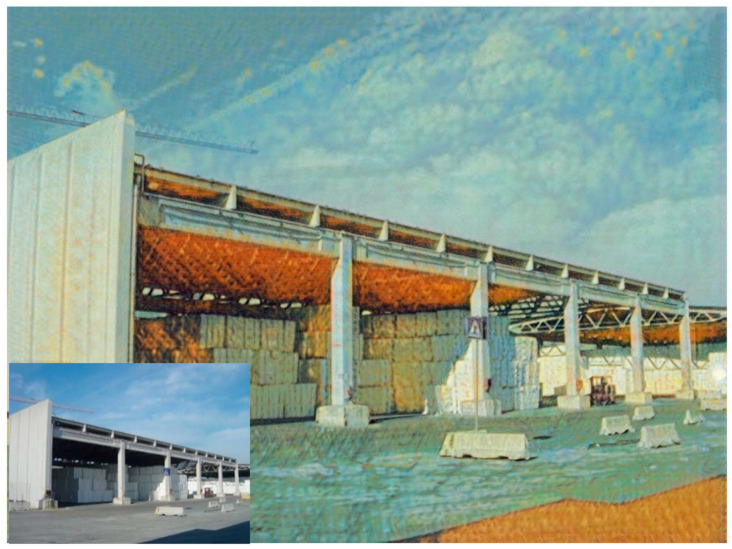
High-resolution image (1280 × 1280 pix). The bottom left corner is the original content image. Even with such a high-definition generated image, we can see a clear thin iron frame.

**Figure 11 sensors-22-06134-f011:**
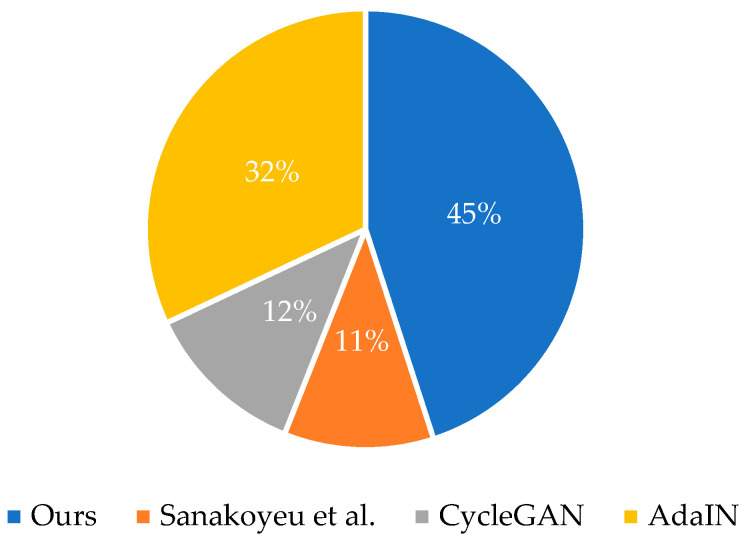
User preference voting for 4 algorithms.

**Figure 12 sensors-22-06134-f012:**
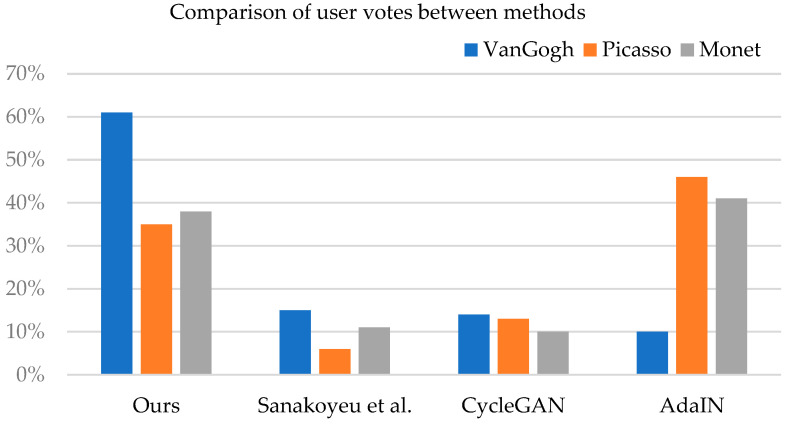
Comparison of user votes between methods.

**Figure 13 sensors-22-06134-f013:**
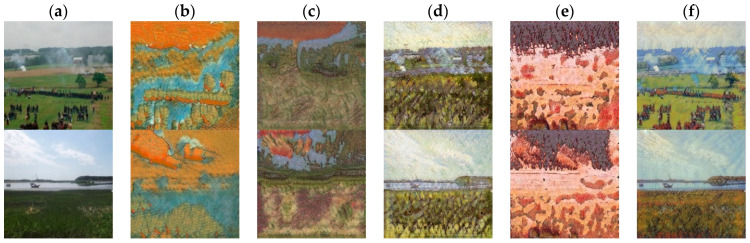
Comparison of the proposed method of ablation experiment. (**a**) Inputs. (**b**) Only $L_{GAN}$. (**c**) Without $L_{c}$. (**d**) Without $L_{s}$. (**e**) Without IFFM. (**f**) Full model.

**Figure 14 sensors-22-06134-f014:**
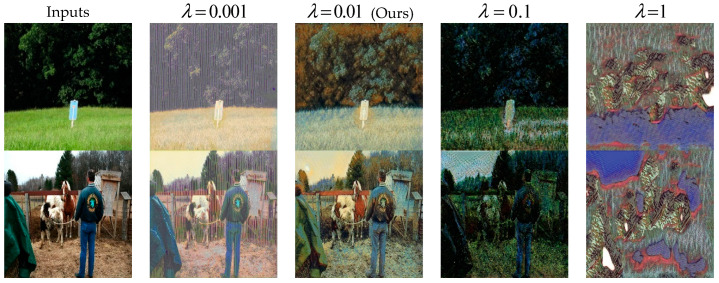
Qualitative comparison of parameter $\lambda =\left\{0.001,\;0.01,\;0.1,\;\left.1\right\}\right.$.

## Data Availability

This work is using public datasets. The content images are from Places365 dataset, and the style images are from Wikiart dataset.

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
