# Peer review of "IFFMStyle: High-Quality Image Style Transfer Using Invalid Feature Filter Modules"

_sensors, 2022, doi:10.3390/s22166134_

Round 1
Reviewer 1 Report
The article presents an improved approach for image style transfer, i.e., given an input image, applying a style of art from some existing artistic method. The authors produced results that were visually superior to existing approaches. The superiority was proved by the users' research when compared with images generated by previous approaches.
The reviewer does not find any issues with the proposed work in view of the related work described in the article.
Could the authors explain the effect of using instance normalization instead of batch normalization in terms of execution performance (the authors claim faster convergence though)? particularly their in regard to their observation in lines 41-42.
Also, the authors did not mention any limitations in their work. I would suggest adding them here.
Here are a few observations:
The first paragraph in 3.1 is template material, it should be removed.
The box symbols on lines 176 and 192 and afterward are unclear. Perhaps a non-unicode symbol? This also implies that the equations/symbols are not well-explained.
Fig 11, the grey color legend is missing.
Reviewer 2 Report
The paper presents a new approach to image style transfer, by incorporating into a GAN-like neural network a new and successful concept, namely, contrastive learning. The paper is well written, all details of the method are well described, and the results are compared to state-of-the-art style transfer methods. (One remark regarding editing: some signs do not appear properly in some equations using the Adobe Reader that I use - see eqs. at line 193 and 206 for instance).
Even though, personally, I tend to favor more results obtained by CycleGAN than those yielded by proposed method, I think the paper deserves publication in Sensors in its current form.
Author Response
Point 1: Some signs do not appear properly in some equations using the Adobe Reader that I use - see eqs. at line 193 and 206 for instance.
Response 1: Regarding the problem that some signs are not displayed, I modified the expression of the formula in the text.
Point 2: I tend to favor more results obtained by CycleGAN than those yielded by proposed method, I think the paper deserves publication in Sensors in its current form.
Response 2: Thank you for your affirmation. I also understand your point of view. Everyone's preferences may be different, so different methods can be used according to different needs.